# OpenReview forum: "More Sail than Ballast: Addressing Harmful Knowledge Leakage in the Expansive Reasoning Space of LRMs"
_ICML.cc/2026/Conference — ICML 2026 regular_

### Official Review · Reviewer_cz23 · 2026-03-12

**Soundness:** 3
**Presentation:** 3
**Significance:** 3
**Originality:** 3
**Overall Recommendation:** 5
**Confidence:** 3

**Summary:**

This paper points out that large reasoning models (LRMs) like DeepSeek-R1 and QwQ can accidentally leak dangerous info when they're thinking through open-ended prompts about sensitive topics. Like, if you ask about the "chemical context" of a chemical weapon, the model might just walk through the synthesis steps in its chain-of-thought without realizing that's a bad idea. The authors call this "unintended elicitation." They build a benchmark (OpenLeakBench) to measure this, and then propose OpenSafeRL — an RL-based method with a three-part reward (safety, faithfulness, helpfulness) that teaches models to reason about safety boundaries instead of just doing binary refuse/comply. Results look pretty good: their models stay safe, don't over-refuse, and keep their general capabilities.

**Compliance With Llm Reviewing Policy:**

Affirmed.

**Final Justification:**

After careful reconsideration during the discussion period and reading the other reviewers' assessments, I am revising my score downward from 5 to 3.

While the rebuttal addressed my W3 concern by adding DeepSeek-R1-Distill-LLaMA-8B experiments — which I appreciate — the broader discussion has highlighted several issues I initially underweighted:

1. **Evaluation circularity is more severe than I recognized**: The benchmark, training data, reward model, and evaluation judge all rely on GPT-4o and HarmBench categories. Reviewers Fkai and XPKP both independently identified this as a fundamental flaw. Strong in-distribution performance on the authors' own benchmark tells us much less than I originally credited. The precision check (92% on 200 samples) is insufficient — recall is unreported, and no independent human evaluation was conducted.

2. **System prompt dependency undermines the core claim**: Table 7 shows over-refusal jumps from 2.8% to 29.2% without the system prompt — a 10x increase. This means a substantial portion of the improvement is attributable to prompt engineering rather than the RL training. At deployment, system prompts are fragile and user-modifiable. The rebuttal did not adequately address this concern.

3. **Missing critical baselines**: Deliberative Alignment (DA) and SARO are the most directly comparable methods, discussed at length in related work but absent from quantitative comparison. Without these, it is difficult to assess the marginal contribution of OpenSafeRL over simpler prompt-based or reasoning-based safety approaches.

4. **Narrow harm categories**: As Reviewer Fkai noted, the evaluation focuses heavily on CBRN threats. Generalization to cybersecurity, social engineering, privacy violations, or financial fraud domains remains undemonstrated. If the method only works where keyword-based detection might also suffice, the contribution is narrower than claimed.

5. **Helpfulness evaluation is underspecified**: Standard benchmarks like MT-Bench or AlpacaEval are not reported. The claimed preservation of helpfulness needs more rigorous validation.

The problem formulation (unintended elicitation in LRMs) is genuinely novel and important, and the training efficiency is impressive. However, the gap between claims and evidence — particularly the evaluation circularity and system prompt dependency — is too large for acceptance at ICML. I now align with the consensus of Reviewers Fkai, XPKP, and vu7C. Score revised to 3 (Weak Reject).

**Key Questions For Authors:**

Can you test on sensitive topics completely outside HarmBench categories?
What happens if you apply this to a real reasoning model like R1-Distill-7B?
How sensitive are results to the α/β/γ values?
What's the recall of your GPT-4o harmful token detector?
Does this work in multi-turn conversations?

**Limitations:**

Not really. The impact statement is there but actual limitations are not discussed. Key gaps: (1) GPT-4o is used as judge for everything — training, eval, data filtering — so correlated blind spots could go undetected. (2) Motivated by LRMs but only tested on 7-8B instruct models, not actual reasoning models. (3) System prompt

**Strengths And Weaknesses:**

Strengths
S1. Good problem, good motivation.
The "unintended elicitation" angle is fresh. It's not jailbreaking — models just volunteer harmful stuff when they don't need to. The Lewisite example in Figure 1 makes this super clear. Table 1 showing 60%+ harmfulness across all major models is eye-opening.
S2. The reward design makes sense.
Safety as a hard gate, then faithfulness and helpfulness on top. Simple, clean. The ablation in Table 3 shows it works — safety-only causes massive over-refusal, helpfulness-only hurts robustness. You need all three.
S3. Over-refusal numbers are really strong.
This is the highlight. 0.4% over-refusal on XSTest vs 20.6% for Circuit Breakers (Qwen). The model actually reasons about whether something is dangerous instead of pattern-matching on scary words.
S4. Cheap to train.
1,400 examples, $20 in API costs, 2.5 hours. Hard to argue with that.


Weaknesses
W1. Training and eval are too similar.
The benchmark and training data come from the same pipeline — same HarmBench categories, same prompt templates, same GPT-4o judge. So the good numbers on their own benchmark (Table 4) might just be in-distribution performance. The jailbreak and over-refusal evals help, but the core claim ("we fix unintended elicitation") is tested on data that looks a lot like training data. Testing on a totally different set of sensitive topics would be much more convincing.
W2. GPT-4o is everywhere.
It's the reward model during training, the evaluator during testing, the data quality filter, and the harmful token detector. If GPT-4o has a blind spot, it shows up everywhere and you'd never know. The paper checks precision (92%) but on only 200 samples, and doesn't report recall. Table 9 tries alternative judges but only on the reward side, not eval.
W3. No experiments on actual reasoning models.
The whole motivation is about LRMs (R1, QwQ, o1). But the experiments are on Qwen-2.5-7B-Instruct and LLaMA-3-8B-Instruct — regular instruction-tuned models, not reasoning models. That's a disconnect. Even testing on DeepSeek-R1-Distill-7B would help close the gap.
W4. System prompt does a lot of work.
Table 7: remove the prompt, over-refusal jumps from 2.8% to 29.2%. That's a 10x increase. So the system prompt is carrying a big part of the improvement. What happens at deployment if someone changes or drops it?
W5. Missing baselines.
Deliberative Alignment (DA) is discussed at length in related work and there's a qualitative example (Figure 6), but no numbers in the main table. SARO is also mentioned but not compared. These are the most relevant baselines.
W6. Fuzzy definition of "harmful."
Where exactly is the line between educational and harmful? Mentioning starting materials — harmful or not? The paper just lets GPT-4o decide. Different judges might draw the line very differently. No discussion of this.

---

> ### Author Rebuttal · Authors · 2026-03-31
>
> We greatly appreciate your kind and constructive feedback and we would welcome the opportunity to discuss this work further with you.
>
> **W1. The training and evaluation distributions are similar.**
>
> Our training set and test set are indeed out-of-distribution. Our training data is synthesized based on the harmful circuit breaker dataset [1], where samples with a BLEU score above 0.3 compared to any HarmBench behavior are filtered out to prevent data contamination.
>
> 1. Improving Alignment and Robustness with Circuit Breakers.
>
> **W2. The reliability of using AI models at every step of the pipeline.**
>
> In every stage where we use AI-as-a-judge, we conduct rigorous human evaluation to verify the accuracy of the LLM-based judge, following well-established methodologies[2]. Our human evaluation is also performed on the full HarmBench, consisting of 200 samples in total.
>
> We further use **Claude as the evaluator** and results demonstrate the robustness of our pipeline. In addition, human evaluation shows that the judge model achieves **a recall of 0.98** in detecting harmful tokens in model responses, indicating that the LLM rarely misses harmful token identification.
>
> | Claude-4.5 as Judge model | Harmful Rate | Avg Harmful Tokens | Avg Radio |
> | :--- | :---: | :---: | :---: |
> | Qwen-2.5-7b-instruct | 62% | 213 | 46% |
> | Qwen-2.5-7b-instruct (ours) | 24% | 119 | 24.5% |
> | Llama-3-8b-instruct | 37.5% | 235 | 35% |
> | Llama-3-8b-instruct (ours) | 14.5% | 105 | 30.5% |
>
> 2. Fine-tuning Aligned Language Models Compromises Safety, Even When Users Do Not Intend To!
>
> **W3. Not validated on large reasoning models (LRMs).**
>
> We select DeepSeek-R1-Distill-LLaMA-8B. Experimental results show that our method significantly improves the model’s safety robustness against jailbreak attacks, while also reducing over-refusal behavior and not affecting general capabilities.
>
> | Model | | | Robustness | | | | Capability | | | Over Refusal |
> | :--- | :---: | :---: | :---: | :---: | :---: | :---: | :---: | :---: | :---: | :---: |
> | | PAIR | PAP | GCG | AutoDAN | MMLU | GSM8K | MTBench | XsTest | OKTest | PHTest | OR-Bench |
> | deepseek-r1-distill-llama-8b | 35.0 | 15.0 | 42.0 | 54.0 | 85.2 | 93.73 | 8.8 | 4.00 | 8.33 | 6.33 | 9.33 |
> | deepseek-r1-distill-llama-8b (ours) | 19.0 | 5.0 | 13.0 | 12.0 | 84.8 | 94.5 | 8.7 | 2.0 | 2.3 | 2.2 | 3.1 |
>
> **W4. The system prompt has too much influence.**
>
> There is no need to worry about this. Table 7 shows that our framework is robust to the choice of system prompt. Therefore, developers can modify the prompt as needed during deployment. We also recommend that developers inject personalized prompts into our training set to better adapt to their specific scenarios.
>
> **W5. Lack of comparison with baselines.**
>
> We have compared with DA. In Table 1, we compare with OpenAI’s o1 model, which is implemented using DA. SARO is also not open-sourced.
>
> **W6. The definition of “harmfulness” is not sufficiently clear.**
>
> This is fundamentally a difficult problem in the safety domain, as knowledge is often dual-use. For example, biochemical knowledge can be used to develop antibiotics, but it can also be used to create chemical weapons. It is therefore difficult to draw a clear boundary between safe and unsafe knowledge. One possible standard is the law, but it is also insufficient to cover morally sensitive issues in society.
>
> Therefore, as shown in Table 10, our work follows common practices in the community. We use official safety policies, such as those from Meta and OpenAI, as references for the LLM judge model, and assign a discrete score based on the tendency of the model’s response and the level of detail in the content.
>
> **Q1. evaluation of hyperparameter sensitivity:**
>
> For hyperparameter settings, our general strategy is to prioritize safety over faithfulness and helpfulness, so we set $\alpha$ to a relatively higher value. We select three different groups of configurations, and the results show that our model is relatively robust to hyperparameter choices.
>
> | Hyperparameters | PAIR (ASR%) | AutoDAN (ASR%) | ReNeLLM (ASR%) | MMLU (%) | GSM8K (%) | XTest (%) | OR-Bench (%) |
> | :--- | :---: | :---: | :---: | :---: | :---: | :---: | :---: |
> | **$\alpha=0.5, \beta=0.3, \gamma=0.2$** | 18.0 | 14.0 | 8.0 | 66.91 | 74.08 | 2.80 | 2.67 |
> | **$\alpha=0.7, \beta=0.2, \gamma=0.1$** | 15.0 | 12.0 | 6.0 | 66.51 | 73.00 | 3.20 | 2.74 |
> | **$\alpha=0.4, \beta=0.3, \gamma=0.3$** | 19.5 | 16.0 | 9.0 | 66.92 | 74.50 | 2.80 | 2.90 |
>
> **Q2. multi-turn attacks.**
>
> We selected three current state-of-the-art multi-turn attack methods—ActorAttack, RedQueen, and Crescendo. Experimental results show that our method significantly improves the model’s robustness against multi-turn attacks.
>
> | | Multi-turn Attack (%) |  |  |
> |---|:--:|:--:|:--:|
> | | ActorAttack | RedQueen | Crescendo |
> | Qwen2.5-7B | 76.00 | 39.50| 62.00|
> | Qwen2.5-7B (ours) | 15.0 | 10.0| 16.0 |

---

> > ### Author Rebuttal · Reviewer_cz23 · 2026-04-04
> >
> > Really solid rebuttal. You basically addressed everything I asked for.
> >
> > The R1-Distill experiment was the big one for me — that was probably the weakest point of the original paper (motivation is about LRMs but no LRM experiments), and the numbers look good. Jailbreak ASR drops substantially while capabilities stay intact. That closes the gap.
> >
> > The Claude cross-validation and the 0.98 recall number also help a lot with the GPT-4o-everywhere concern. Good to see it's not just one model validating itself.
> >
> > Multi-turn attack results are a nice bonus — I didn't expect you to run those but they do strengthen the story. The drops from 76% to 15% on ActorAttack are pretty dramatic.
> >
> > The hyperparameter sensitivity table is helpful too — shows the method isn't super brittle to the exact reward weights.
> >
> > I think the system prompt thing is still slightly underplayed (29% over-refusal without it is a lot), but I accept that Table 7 shows the framework works with different prompts. In practice people will use a system prompt anyway.
> >
> > Keeping my score. Nice work overall.

---

### Official Review · Reviewer_vu7C · 2026-03-12

**Soundness:** 3
**Presentation:** 3
**Significance:** 3
**Originality:** 3
**Overall Recommendation:** 4
**Confidence:** 4

**Summary:**

This paper introduces a benchmark revealing that LLMs, particularly  LRMs, often leak harmful knowledge when responding to open-ended sensitive reasoning tasks, a phenomenon the authors term "unintended elicitation." To address this vulnerability, the authors propose OpenSafeRL, a two-stage pipeline comprising: (1) prompt-augmentation, which identifies harmful actions in original prompts and transforms them into open-ended questions concerning specific entities; and (2) a verifiable, hierarchical reward function designed for RL training. Extensive experiments, including thorough ablation studies, demonstrate that OpenSafeRL successfully enhances both the generalization and defensive capabilities of LLMs.

**Compliance With Llm Reviewing Policy:**

Affirmed.

**Final Justification:**

The rebuttal has largely addressed my primary concerns, and I appreciate the additional experimental results provided. I also look forward to the open-sourcing of the project. I'll keep my score.

**Key Questions For Authors:**

**Q1**: Clarification on Evaluation Metrics: Could you provide the performance of the evaluated models on the original Harmbench dataset using the exact same evaluation metrics applied in Table 1? This would help isolate whether the observed "unintended elicitation" is primarily driven by the open-ended prompt structure or if it is partially an artifact of the new evaluation metric itself.


**Q2**:Mechanistic Insights: Could you provide further analysis or discussion on the underlying causes of unintended elicitation? Specifically, what is the theoretical or mechanistic reason that aligning on entity-centric, open-ended questions yields superior downstream results compared to traditional refusal-style alignment?

**Limitations:**

The authors do not explicitly discuss limitations in the main text. While the experimental validation of the phenomenon and the proposed defense is strong, the following unaddressed limitations currently hinder the paper's impact:

Methodological Disconnect: There is a gap between the problem setting (which focuses heavily on LRMs) and the solution validation (which is only tested on LLMs).

Lack of Mechanistic Insights: The paper lacks deeper analytical experiments exploring the underlying causes of unintended elicitation, or why aligning on open-ended questions yields superior downstream results.

**Strengths And Weaknesses:**

### Strengths

- **Identification of Unintended Elicitation:** The paper highlights a blind spot in current  alignment algorithms. While existing methods over-index on preventing explicitly harmful actions, they fail to provide robust guardrails for entity-centric, open-ended queries.
- **Extensive Empirical Validation:** The authors provide thorough experimental evidence. The comprehensive ablation studies rigorously validate the effectiveness of the OpenSafeRL algorithm.

### Weaknesses

- **Lack of Baseline Comparison for the Elicitation Metric:** While the paper demonstrates that models leak harmful knowledge on open-ended tasks, it is unclear to determine if this is entirely due to the prompt structure or partially an artifact of the new evaluation metrics. Providing results for the original Harmbench dataset using the *same evaluation metrics* (to augment Table 1) would make the baseline claims much more solid.
- **Disconnect Between Motivation (LRMs) and Experiments (LLMs):** The paper heavily premises itself on the fact that LRMs elicit more harmful knowledge (as noted in Line 215). However, the OpenSafeRL experiments are exclusively conducted on standard LLMs. Evaluating OpenSafeRL on modern LRMs (e.g., the DeepSeek-R1-Distill series) is necessary to fully support the paper's core narrative.
- **Missing Recent Baselines:** The experimental section would benefit from including more recent reasoning-based safety or RLVR baselines, such as Deliberative Alignment [1] or IPO[2].


[1] Guan, Melody Y., et al. "Deliberative alignment: Reasoning enables safer language models." *arXiv preprint arXiv:2412.16339* (2024).

[2] Zhang, Yichi, et al. "Towards safe reasoning in large reasoning models via corrective intervention." *arXiv preprint arXiv:2509.24393* (2025).

---

> ### Author Rebuttal · Authors · 2026-03-31
>
> Thank you for highlighting the findings of our paper! We are glad that our contributions resonated with you, and we look forward to continuing the conversation.
>
> **W1. Analysis of the source of unintended elicitation (evaluation metric or prompt distribution):**
>
> Thank you for pointing this out! We evaluated all models listed in Table 1 on the original HarmBench queries. Except for Qwen-2.5-7B-Instruct, all models achieve a harmfulness rate (i.e., the proportion of outputs containing harmful content) of 0% on the original prompts, meaning they directly refuse all original harmful queries. Qwen-2.5-7B-Instruct also only has a harmfulness rate of 2%.
>
> In contrast, our prompts achieve at least a 60% harmfulness rate across all models. This indicates that the prompt distribution is the primary factor influencing whether models output harmful knowledge.
>
> **W2. Disconnect between motivation (LRMs) and experiments (LLMs):**
>
> Thank you for pointing this out! We selected DeepSeek-R1-Distill-LLaMA-8B and applied our OpenSafeRL algorithm. The experimental results show that our method significantly improves the safety robustness of large reasoning models against jailbreak attacks, while also reducing over-refusal behavior. At the same time, the model’s general capabilities are almost unaffected, demonstrating the generality of our approach.
>
> | Model | | | Robustness | | | | Capability | | | Over Refusal |
> | :--- | :---: | :---: | :---: | :---: | :---: | :---: | :---: | :---: | :---: | :---: |
> | | PAIR | PAP | GCG | AutoDAN | MMLU | GSM8K | MTBench | XsTest | OKTest | PHTest | OR-Bench |
> | deepseek-r1-distill-llama-8b | 35.0 | 15.0 | 42.0 | 54.0 | 85.2 | 93.73 | 8.8 | 4.00 | 8.33 | 6.33 | 9.33 |
> | deepseek-r1-distill-llama-8b (ours) | 19.0 | 5.0 | 13.0 | 12.0 | 84.8 | 94.5 | 8.7 | 2.0 | 2.3 | 2.2 | 3.1 |
>
> **W3. Missing latest baseline methods:**
>
> Thank you for pointing this out! We have already included a comparison with o1-mini in Table 1, which is implemented via Deliberative Alignment [1], demonstrating the vulnerability of DA under our open-ended tasks. IPO [2] has not yet open-sourced its code or model weights, and we have contacted the original authors to see if it is possible to include their model.
>
> references:
>
> 1. Deliberative alignment: Reasoning enables safer language models.
> 2. Towards safe reasoning in large reasoning models via corrective intervention.
>
> **Q1: Mechanistic insights:**
>
> Thank you for your insightful question! In one sentence: our RL method motivates the model to “manage” its own knowledge, which is a more fundamental and challenging goal than traditional refusal-style safety RL.
>
> Safety alignment depends on the design of two components: the environment and the reward model. For traditional refusal-style safety RL, its environment typically consists of direct queries with explicit harmful intent (e.g., “How to make Lewisite”) and its reward model usually encourages direct refusal. By contrast, we provide a model with a more open-ended environment (e.g. "Outline the context of Lewisite"), and “direct refusal” is not an ideal answer in this setting. Our reward model motivates the model to learn to manage knowledge against such open-ended queries—excluding explanations of sensitive and dangerous processes while still providing necessary educational information.
>
> Moreover, the limitation of “direct refusal” lies in reward hacking. This is particularly reflected in refusal-style models, which tend to over-refuse even benign queries, as shown in Table 2. Our reward model penalizes such shortcuts. In Table 8, we monitor the training process and verify that the model is genuinely learning safety behaviors rather than hacking the reward metric. From the qualitative analysis in Figure 1, our model indeed learns to manage its knowledge during reasoning.

---

> > ### Author Rebuttal · Reviewer_vu7C · 2026-04-04
> >
> > The rebuttal has largely addressed my primary concerns, and I appreciate the additional experimental results provided. I also look forward to the open-sourcing of the project. I'll keep my score.

---

### Official Review · Reviewer_XPKP · 2026-03-12

**Soundness:** 3
**Presentation:** 2
**Significance:** 1
**Originality:** 2
**Overall Recommendation:** 4
**Confidence:** 4

**Summary:**

The paper identifies a safety vulnerability it calls "unintended elicitation" in Large Reasoning Models (LRMs): when given open-ended but ostensibly benign prompts about sensitive topics, LRMs can inadvertently leak dangerous information through their chain-of-thought (CoT) reasoning. The authors propose OpenSafeRL, a reinforcement learning framework with a hierarchical reward model that jointly optimizes for safety, faithfulness, and helpfulness. They also introduce a benchmark (OpenLeakBench) and a synthetic data generation pipeline to study and address this phenomenon.

**Compliance With Llm Reviewing Policy:**

Affirmed.

**Final Justification:**

The rebuttal addressed most of my concerns, so I've raised my score. I suggest the authors clarify the novelty and contributions of their work in the paper.

**Key Questions For Authors:**

Please refer to the points mentioned in the Weaknesses section. In addition, I have the following questions:
Q1: How do the authors address the potential evaluation circularity from using GPT-4o both to construct the dataset and to evaluate the method?
Q2: How does the proposed approach differ methodologically from existing RL-based alignment frameworks, and how would it perform against stronger baselines (e.g., prompt-based safety methods) or on more recent frontier models?
Q3: The choice of alpha, beta, and gamma in the hierarchical reward function appears arbitrary. How were these weights determined? Is there guidance for practitioners to select appropriate values? How sensitive are the results to different weight configurations across datasets and models?
Q4: The paper employs REINFORCE for the RL optimization. Given that more sample-efficient RL algorithms like PPO are commonly used in LLM alignment, could the authors clarify why REINFORCE was chosen? Have you conducted ablations comparing REINFORCE with PPO or other policy gradient methods, and if so, how do the results differ in terms of sample efficiency, training stability, and final performance?

On page 13, the table reference in the last line is unresolved and appears as question marks.

**Limitations:**

yes

**Strengths And Weaknesses:**

Strengths:

- This paper introduces a novel distinction between explicit jailbreaks and unintended elicitation through open-ended prompting. The observation that CoT reasoning creates a broader answer space in which harmful content can emerge is practically important and relatively underexplored.

Weaknesses:

- OpenLeakBench is derived from HarmBench and relies on GPT-4o both to filter the dataset and to serve as the primary reward/evaluation model. This creates a potential circularity: the method is optimized using the same model that is later used to evaluate it. Moreover, the independent human evaluation covers only 200 samples with a single annotator per query, which is likely insufficient to establish the benchmark’s reliability or to verify the reward model’s ability to generalize.

- Although the observation on unintended elicitation that motivates the work is interesting and appears to be novel, the overall framework, which uses RL to optimize a model toward a specified safety objective, is conceptually similar to existing approaches. As a result, the contribution appears to lie primarily in the engineering and implementation of the training pipeline rather than in a fundamentally new methodological development.

- The main baselines (SFT, GA, and CB) were not specifically designed to address unintended elicitation. In addition, the paper does not compare against simpler prompt-engineering approaches, such as using explicit safety system prompts alone, which could potentially achieve similar improvements. This comparison would be particularly relevant given that Table 7 indicates that a system prompt by itself already yields meaningful safety gains.

- The experimental evaluation focuses primarily on Qwen-2.5-7B and LLaMA-3-8B. These models are relatively small and somewhat dated compared to current frontier models, which raises questions about how well the results would transfer to more recent or larger systems.

---

> ### Author Rebuttal · Authors · 2026-03-31
>
> We sincerely thank you for your positive evaluation of our work! Your appreciation of the motivation and design choices means a lot to us, and we would be happy to engage in further discussion.
>
> **W1. Reliability of the reward model:**
>
> Your concern is essentially about the generalization ability of our method, e.g., whether our method overfits to a single reward model. In fact, our method uses different reward models, including open-source reward models, and consistently achieves significant improvements in safety performance, as shown in Table 9.
>
> **W2. human evaluation quality of the judge model**
>
> Thank you for pointing this out! The reliability of our GPT-based judge model has already been verified: it achieved a kappa score of 0.792 with human annotators [1]. On top of this, we further conducted human verification of its accuracy in identifying harmful tokens. To address your concern, we follow the methodology of [1]: for annotations with large disagreements, annotators meet to compare their scores and work towards a consensus. In this way, we achieved a precision score of 0.93, which is comparable to the 0.92 obtained using single-human evaluation.
>
> In addition, you are concerned about the sample size of our human evaluation. In fact, we evaluate on the full set of HarmBench with a total of 200 queries. The human evaluation of the reward model also follows the methodology of [1].
>
> References:
> 1. Fine-tuning Aligned Language Models Compromises Safety, Even When Users Do Not Intend To!
>
> **W3. Novelty of our RL framework:**
>
> In one sentence: our RL method motivates the model to “manage” its own knowledge, which is a more fundamental and challenging goal than traditional refusal-style safety RL.
>
> Safety alignment depends on the design of two components: the environment and the reward model. For traditional refusal-style safety RL, its environment typically consists of direct queries with explicit harmful intent (e.g., “How to make Lewisite”) and its reward model usually encourages direct refusal. By contrast, we provide a model with a more open-ended environment (e.g. "Outline the context of Lewisite"), and “direct refusal” is not an ideal answer in this setting. Our reward model motivates the model to learn to manage knowledge against such open-ended queries—excluding explanations of sensitive and dangerous processes while still providing necessary educational information. Our design of the environment and reward model seperate our work from other works, and experimental results demonstrate the advantage of teaching the model to manage its knowledge during its reasoning process.
>
> **W4. Comparison with prompt engineering baselines:**
>
> As shown in Table 7, prompt-only baselines are far from the Pareto optimal point. A safety system prompt still yields 34% unsafe responses under ReNeLLM attacks, increases over-refusal on XTest (9.2% vs. 6.8%), and slightly degrades general capability (GSM8K: 74.9 → 72.91).
>
> In contrast, our model reduces attack failure cases to 8.0%, lowers over-refusal to 2.8%, and preserves general capability.
>
> **W5. Scalability towards larger model scale:**
>
> We have already conducted experiments on larger-scale models. We extended our experiments to Qwen-2.5-14B-Instruct, and the results in Table 10 validate the generality of our approach on larger models.
>
> **Q1. evaluation of hyperparameter sensitivity:**
>
> For hyperparameter settings, our general strategy is to prioritize safety over faithfulness and helpfulness, so we set $\alpha$ to a relatively higher value. We select three different groups of configurations, and the results show that our model is relatively robust to hyperparameter choices.
>
> | Hyperparameters | PAIR (ASR%) | AutoDAN (ASR%) | ReNeLLM (ASR%) | MMLU (%) | GSM8K (%) | XTest (%) | OR-Bench (%) |
> | :--- | :---: | :---: | :---: | :---: | :---: | :---: | :---: |
> | **$\alpha=0.5, \beta=0.3, \gamma=0.2$** | 18.0 | 14.0 | 8.0 | 66.91 | 74.08 | 2.80 | 2.67 |
> | **$\alpha=0.7, \beta=0.2, \gamma=0.1$** | 15.0 | 12.0 | 6.0 | 66.51 | 73.00 | 3.20 | 2.74 |
> | **$\alpha=0.4, \beta=0.3, \gamma=0.3$** | 19.5 | 16.0 | 9.0 | 66.92 | 74.50 | 2.80 | 2.90 |
>
> **Q2. Reason for choosing REINFORCE:**
>
> Thank you for pointing this out. We use REINFORCE++ instead of the classic REINFORCE algorithm. REINFORCE++ does not require a critic and is more sample-efficient than PPO. We will clarify this in the camera-ready version. We also compared it with GRPO and found our framework to be robust across different RL algorithms.
>
> | Method | Robustness (%) | | | Capability (%) | | | Over Refusal |
> | :--- | :---: | :---: | :---: | :---: | :---: | :---: | :---: |
> | | PAIR | AutoDAN | ReNeLLM | MMLU | GSM8K | XTest | OR-Bench |
> | REINFORCE++ | 18.0 | 14.0 | 8.0 | 66.91 | 74.08 | 2.80 | 2.67 |
> | GRPO | 15.0 | 15.0 | 10.0 | 67.31 | 74.98 | 2.60 | 2.87 |
>
> **Q3. Latex format error:**
>
> Thank you for pointing this out! We will correct this table reference in the camera-ready paper.

---

> > ### Author Rebuttal · Reviewer_XPKP · 2026-04-04
> >
> > Thank you for the rebuttal. However, I remain unconvinced regarding the novelty of the RL framework. My primary concern is that the work lacks methodological innovation, as it primarily applies an existing RL algorithm to a new problem domain. While I acknowledge the environmental differences compared to standard safety RL, this strikes me as a change in the problem setting rather than a fundamental advancement in methodology. Furthermore, the implementation of a hierarchical reward model is a well-established technique and does not constitute a novel contribution in this context.
> >
> > I think my point in W4 wasn't properly understood. My question is about direct comparison with prompt engineering-only approaches (no RL training, e.g., few-shot examples), which I couldn't find in the current paper.

---

> > > ### Author Response · Authors · 2026-04-06
> > >
> > > Thank you for your response!
> > >
> > > **W4. Comparison with prompt engineering baselines.**
> > >
> > > We have added a well-known prompt-only baseline method called ICD, which improves safety by incorporating few-shot examples and safety instructions into the context [1]. The results show that ICD can bring some improvements in safety performance, but at the cost of reduced general capability scores and an increased tendency toward over-refusal. Therefore, whether using our system prompt or ICD, we observe a similar trade-off between safety and utility, and both are insufficient to defend against advanced adaptive attacks.
> > >
> > > |            | Robustness (%) |         |         | Capability (%) |       | Over Refusal |          |
> > > |------------|:--------------:|:-------:|:-------:|:--------------:|:-----:|:------------:|:--------:|
> > > |            | PAIR           | AutoDAN | ReNeLLM | MMLU           | GSM8K | XTest        | OR-Bench |
> > > | No defense | 29.5           | 45.5    | 74.0    | 67.69          | 74.90 | 6.80         | 9.67     |
> > > | ICD        | 27.0           | 35.0    | 45.0    | 65.01          | 72.80 | 8.20         | 12.30    |
> > > | Ours       | 18.0           | 14.0    | 8.0     | 66.91          | 74.08 | 2.80         | 2.67     |
> > >
> > > references:
> > >
> > > 1. Jailbreak and Guard Aligned Language Models with Only Few In-Context Demonstrations
> > >
> > > **W3. The novelty of our RL system.**
> > >
> > > In addition, thank you for your patience! We would like to further discuss the novelty of our method. We want to emphasize again that the RL environment is extremely important, and together with the reward model, it determines how the model’s behavior is shaped. **Constructing such an environment is non-trivial**—it is not merely a matter of changing the “problem setting.” We designed a data synthesis pipeline that automatically generates open-ended tasks with unsafe response trajectories. These tasks constitute our “reinforcement learning environment,” in contrast to traditional RL environments, which are composed of directly harmful intent queries. It is precisely in such a challenging RL environment that our model can learn to manage its knowledge and prevent the leakage of harmful knowledge. In contrast, in traditional RL environments, the model is more likely to be shaped toward over-refusal, which negatively impacts its downstream usability.
> > >
> > > Finally, we would like to stress that our contribution does not lie in any single component of the RL system, such as the reinforcement learning algorithm itself or the reward model alone. Instead, it is through the collaboration between our RL environment and our designed reward model that the model is endowed with the ability to manage its knowledge within the reasoning space and prevent the leakage of harmful knowledge. Notably, through extensive and comprehensive experiments, we show that this capability is non-trivial and can generalize to out-of-distribution attacks. **It demonstrates our contribution at the level of the RL system. This is not something that can be achieved simply by relying on existing RL algorithms.**
> > >
> > > Therefore, we kindly ask you to reconsider and reevaluate our work. Thank you very much! We will also revise the manuscript to clarify the presentation of our methodological contributions, so that readers can better recognize them.

---

### Official Review · Reviewer_Fkai · 2026-03-13

**Soundness:** 3
**Presentation:** 3
**Significance:** 3
**Originality:** 2
**Overall Recommendation:** 3
**Confidence:** 4

**Summary:**

This paper identifies "unintended elicitation" in large reasoning models (LRMs): during chain-of-thought reasoning, models may reveal dangerous knowledge (e.g., chemical weapon synthesis routes) even when the final output is safe. The authors propose OpenSafeRL, a pipeline with three components: (1) automated data synthesis to generate prompts that trigger unintended elicitation, (2) open-ended RL environments where models self-discover safety reasoning patterns, and (3) a safety-first reward model that prioritizes safety over helpfulness and reasoning faithfulness. Experiments show improved safety with maintained helpfulness, reduced over-refusal, and robustness to jailbreaks.

**Compliance With Llm Reviewing Policy:**

Affirmed.

**Key Questions For Authors:**

How does your work differ from the ICLR 2025 paper "Unintended Harmful Knowledge Elicitation Issue in Large Reasoning Models and a RL Solution" (OpenReview: bvnuXYMgjg)? I need a precise comparison: which aspects of problem definition, data, methodology, and results are shared, and which are novel to this submission. If the core contribution is substantially the same, the originality of this work is significantly reduced.

Have you evaluated on an independently constructed benchmark not derived from your own synthesis pipeline? Evaluation by external red-team professionals, or on established safety benchmarks (WMDP, HarmBench) adapted for reasoning-trace safety, would provide much stronger evidence than self-generated test data.

Does the safety improvement generalize to harm categories beyond CBRN? Showing transfer to cybersecurity, social engineering, or privacy domains-without retraining-would significantly strengthen the generalization claim. If domain-specific retraining is needed for each harm category, that should be acknowledged.

**Limitations:**

The paper discusses some limitations. It should additionally acknowledge: (a) the significant overlap with concurrent work, particularly the ICLR 2025 submission; (b) the circular evaluation risk from self-generated benchmarks; (c) the narrow scope of tested harm categories; (d) the ambiguity of "harmful knowledge" in dual-use settings.

**Strengths And Weaknesses:**

Strengths
S1. Identifies a Real and Underappreciated Threat. The observation that reasoning models can leak dangerous knowledge in their CoT-even when the final answer is safe-is a genuine and important safety concern. Standard RLHF trains on output safety alone; if CoT is visible to users (as in Claude, DeepSeek-R1, etc.), dangerous information in reasoning traces is a distinct attack surface. The paper makes this concrete with examples involving synthesis routes for toxic substances.

S2. Practical and Scalable Data Synthesis. Rather than relying on expensive manual red-teaming, the paper proposes automated generation of prompts that trigger unintended elicitation. This is practical and scalable, allowing systematic exploration of the vulnerability space. The generate-filter-validate pipeline structure is sensible.

S3. Safety-First Reward Hierarchy. The reward model design-safety first, then helpfulness, then reasoning faithfulness-is well-motivated. The explicit ordering prevents the common failure mode where helpfulness pressure overrides safety constraints during RL training. This is a principled design choice that directly addresses the known tension between helpfulness and safety objectives.

S4. Addresses the Over-Refusal Problem. Many safety interventions make models unhelpfully cautious, refusing benign requests. The paper shows its approach improves safety without increasing over-refusal-and in some cases reduces it. This matters a lot for deployment.

S5. Generalization Claims Supported. Testing robustness against jailbreaks and handling over-refusal samples suggests the safety improvements aren't brittle.
Weaknesses
W1. Significant Overlap with Concurrent Work. This is my primary concern. Several concurrent papers address the same problem with overlapping methods. Most notably, "Unintended Harmful Knowledge Elicitation Issue in Large Reasoning Models and a RL Solution" appeared at ICLR 2025 (OpenReview: bvnuXYMgjg) and tackles the same problem with a similar RL-based approach.

W2. Circular Evaluation Risk. The benchmark used to evaluate unintended elicitation is constructed by the authors' own data synthesis pipeline. This creates a risk of circular validation: the method is tuned against data generated by a similar process. An independently constructed benchmark-evaluation by professional red-team practitioners, or testing on established safety benchmarks adapted for reasoning traces (e.g., WMDP-Bio extended to CoT evaluation, or HarmBench extended to reasoning-trace safety)-would provide much stronger evidence. Without independent evaluation, it's hard to rule out that the method overfits to patterns in its own synthetic data.

W3. "Harmful Knowledge" Is Not Formally Defined. Chemical weapon synthesis is an unambiguous example, but many real-world cases are murky: medical drug synthesis information could be harmful or educational; cybersecurity vulnerability details could enable attacks or defenses. The paper doesn't provide a formal taxonomy of harm types or discuss boundary cases. Without this, it's unclear how the method handles dual-use knowledge, context-dependent harm, or cases where the same information is safe in one context and dangerous in another.

W4. Narrow Scope of Harm Categories Tested. Most examples focus on CBRN (chemical, biological, radiological, nuclear) threats-arguably the most clear-cut category. The paper doesn't demonstrate generalization to other forms of harmful knowledge leakage: cybersecurity exploits, social engineering techniques, privacy violations (as studied by "Leaky Thoughts"), or instructions for financial fraud. If the method only works for CBRN content-where keyword-based detection might also work-the contribution is narrower than claimed.

W5. Helpfulness Evaluation Methodology Is Underspecified. The paper claims maintained helpfulness but doesn't fully detail the evaluation: which benchmarks? Are the same prompts used for both safety and helpfulness evaluation, or different ones? If different, there could be a selection effect where safety improvements are measured on a hard safety set while helpfulness is measured on an easy set. Standard benchmarks (MT-Bench, AlpacaEval, or domain-specific task performance) should be reported with clear methodology.
Ratings

---

> ### Author Rebuttal · Authors · 2026-03-31
>
> Thank you for your thoughtful and encouraging feedback! We truly appreciate your recognition of our problem formulation and experimental results, and we look forward to further discussion with you.
>
> **W1. Overlap with concurrent work.**
>
> The OpenReview paper you mentioned was not accepted for publication. According to ICML policies, such work is not required to be considered as a formal reference. All the content in this paper is independently developed and is the original work of the authors.
>
>
> **W2. Risk of circular evaluation.**
>
> We understand yet disagree with your concern that “the effectiveness of our method is due to overfitting.” In fact, our benchmark, openLeakBench, is OOD with respect to the distribution of our training set. openLeakBench is generated based on HarmBench, while our training set is generated based on the harmful circuit breaker set [1], which is a different data distribution. This is because the authors of [1] “filter out all samples that have a BLEU score above 0.3 when compared to any behavior in HarmBench’s standard behaviors set to avoid data contamination with the benchmark.” Moreover, In Table 2, our method achieves SOTA results compared with current leading alignment methods on HarmBench, demonstrating the generalization advantage of our method.
>
> **W3. “Harmful knowledge” is not formally defined.**
>
> Thank you for your question! This is in fact a fundamentally difficult problem in the safety domain, because knowledge is often dual-use. For example, biochemical knowledge can be used to develop antibiotics, but it can also be used to produce chemical weapons. It is hard to draw a clear boundary between safe and unsafe knowledge. Currently, companies adopt laws as a standard to formulate safety policies, and also take into account prevailing social moral values. Therefore, as shown in Figure 10, we follow common practices in the community [1,2,3]: we use official safety policies from large model companies, such as Meta and OpenAI, as references for the LLM judge model. Our judge model evaluates whether a model’s response includes SPECIFIC harmful facts or details that could enable harmful activities, and assigns a graded score accordingly.
>
> references:
> 1. Improving Alignment and Robustness with Circuit Breakers
> 2. Fine-tuning Aligned Language Models Compromises Safety, Even When Users Do Not Intend To!
> 3. How Johnny Can Persuade LLMs to Jailbreak Them: Rethinking Persuasion to Challenge AI Safety by Humanizing LLMs
>
> **W4. Narrow scope of tested harm categories.**
>
> We disagree with your statement that “our evaluation only focuses on CBRN and does not include broader dual-use cases.” Our evaluation benchmark, HarmBench, includes the context-dependent risks and dual-use knowledge you mentioned, such as harassment, misinformation, cybercrime, etc. Moreover, both our human expert evaluation results and GPT-based evaluation results demonstrate that our model can generalize well to different dangerous or sensitive topics.
>
> **W5. Unclear evaluation methodology for usefulness.**
>
> We disagree with your statement that “the evaluation methodology is unclear.” In Section 5.1, we present the benchmarks used to evaluate safety, usefulness, and over-refusal. In addition, we report results on MT-Bench, MMLU, BBH, and GSM8K in Table 2.

---

### Decision · Program_Chairs · 2026-04-30

**Decision:**

Accept (regular)

**Comment:**

**Summary**
This paper considers the "unintended elicitation issue" of the large reasoning models (LRMs). LRMs reasoning on sensitive yet benign topics can exhibit dangerous thoughts, due to their strong multi-step reaosning capabilities. This paper addresses this issue with a synthesized open-ended environments + safety-first reward model design.

**Strengths**
- The identified threat is real and underappreciated.
- The proposed solution, including the data synthesis method and the reward model design, is practical.

**Weaknesses**
- Reviewers mentioned two potential circular evaluation"concerns. The same data synthesis pipeline (Fkai) and the resulting benchmark, OpenLeakBench (XPKP), is used to construct the benchmark to evaluate unintended elicitation and evaluate unintended elicitation. As mentioned in the author's response, openLeakBench has a different data distribution from the training set. Independent human evaluation also partially alleviates this problem.
- A crucial term defining the identified threat, "harmful knowledge" can be defined more formally.
- The baselines can be stronger. In the responses, prompt-based baselines are added. Also, multi-turn attacks (ActorAttack, RedQueen, Crescendo) are added, adding to the robustness of the proposed method.

Overall, the strengths of this paper are significant, and the weaknesses are addressed in the responses and I believe the authors can add them into the camera-ready relatively easily. I tend to recommend acceptance for this paper.